# Genomic assessment of quarantine measures to prevent SARS-CoV-2 importation and transmission

Dinesh Aggarwal [1,2,3,4✉], Andrew J. Page [5,104], Ulf Schaefer[2,104], George M. Savva[5], Richard Myers[2], Erik Volz [6], Nicholas Ellaby[2], Steven Platt[2], Natalie Groves[2], Eileen Gallagher[2], Niamh M. Tumelty [7], Thanh Le Viet [5], Gareth J. Hughes [8], Cong Chen[2], Charlie Turner[2], Sophie Logan[9], Abbie Harrison [2], The COVID-19 Genomics UK (COG-UK) Consortium*, Sharon J. Peacock [1,2,3,4], Meera Chand[2,105] & Ewan M. Harrison[1,2,4,10,105✉]

Mitigation of SARS-CoV-2 transmission from international travel is a priority. We evaluated the effectiveness of travellers being required to quarantine for 14-days on return to England in Summer 2020. We identified 4,207 travel-related SARS-CoV-2 cases and their contacts, and identified 827 associated SARS-CoV-2 genomes. Overall, quarantine was associated with a lower rate of contacts, and the impact of quarantine was greatest in the 16–20 age-group. 186 SARS-CoV-2 genomes were sufficiently unique to identify travel-related clusters. Fewer genomically-linked cases were observed for index cases who returned from countries with quarantine requirement compared to countries with no quarantine requirement. This difference was explained by fewer importation events per identified genome for these cases, as opposed to fewer onward contacts per case. Overall, our study demonstrates that a 14-day quarantine period reduces, but does not completely eliminate, the onward transmission of imported cases, mainly by dissuading travel to countries with a quarantine requirement.

[1] University of Cambridge, Department of Medicine, Cambridge, UK. [2] Public Health England, 61 Colindale Ave, London NW9 5EQ, UK. [3] Cambridge University Hospital NHS Foundation Trust, Cambridge, UK. [4] Wellcome Sanger Institute, Hinxton, Cambridge, UK. [5] Quadram Institute Bioscience, Norwich Research Park, Norwich NR4 7UQ, UK. [6] Imperial College London, Department of Infectious Disease Epidemiology, London, UK. [7] University of Cambridge, Cambridge University Libraries, Cambridge, UK. [8] Public Health England National Infections Service, Field Service, Leeds, UK. [9] Public Health England, National Infections Service, Field Service, Nottingham, UK. [10] University of Cambridge, Department of Public Health and Primary Care, Cambridge, UK. [104] These authors contributed equally: Andrew J. Page, Ulf Schaefer. [105] These authors jointly supervised this work: Meera Chand, Ewan M. Harrison. *A list of authors and their affiliations appears at the end of the paper. ✉email: dinesh.aggarwal@nhs.net; eh6@sanger.ac.uk

SARS-CoV-2 was first identified in Wuhan, China[1] in December 2019 and has since been imported into virtually every country and region in the world[2,3]. Understanding and tracking the sources of importation between countries can give important information for policymakers, and for managing the pandemic, by informing policies aimed at reducing the further spread of virus[4]. It is particularly important now as countries aim to mitigate the introduction of highly transmissible variants of concern with potentially reduced vaccine efficacy[5,6]. The available brakes on imported SARS-CoV-2 cases include travel bans, quarantine measures, and testing of returning travellers[7]. These can apply to all countries or targeted to high-risk countries, for variable durations, and with variable degrees of enforcement.

In England, from 17 March 2020 to 4 July 2020, the government advised against all non-essential travel worldwide[8]. Between 4 July 2020 and 1 February 2021, travel corridors to countries deemed to be low risk for COVID-19 disease (subject to assessment and change) were established in which returning travellers were no longer required to quarantine for 14 days (at home). Persons returning from countries outside this list (except for exemptions e.g. specific employment) were required to quarantine at home (Fig. 1). This policy aimed to reduce the impact of travel-related SARS-CoV-2 cases in England[9] possibly through limiting onwards transmission of SARS-CoV-2[10] and deterring travel to those countries. Upon identification of an imported case, contact tracing and quarantine/self-isolation measures can limit onwards transmission[11]. The PHE Isolation Assurance Service identified up to 97% self-reported compliance with travel-specific quarantine[12]. These data do not include countries exempt from quarantine, contact-tracing data or link to genomic data to evaluate travel-related clusters.

Studies from numerous countries have used genome sequencing to complement epidemiological investigations in order to characterise importations of SARS-CoV-2 (Supplementary Table 1). Primarily these are in-depth case reports on small datasets but demonstrate the utility of genomics combined with contact tracing[13–35]. Here, we combine contact-tracing data from National Health Service (NHS) Test and Trace (T&T) for probable importation cases with genomes made available through the COVID-19 Genomics UK (COG-UK) consortium[36] to characterise the known imported cases of SARS-CoV-2 into England and the effectiveness of 14-day quarantine on onwards transmission.

In this work, we compare the number of contacts reported per case prior to diagnosis between individuals returning from a country with a requirement to quarantine after travel to those who did not need to quarantine on return. We then identify unique genomes from imported cases and associated clusters of infections in the COG-UK genomic surveillance dataset in the four weeks following index case identification. This onward transmission is compared when the index case returned from a country with a requirement to quarantine on return compared to countries without a requirement. Finally, we use the epidemiological data to investigate the origin of a divergent cluster of SARS-CoV-2 cases identified using genomics.

## Results

**Cases identified**. Between 27 May 2020 and 13 September 2020, using contact-tracing data for all individuals who had tested positive for SARS-CoV-2 between those dates, we identified 4207 international travel-related cases in England. These individuals reported a total of 18,856 contacts. During this period, we identified 105,794 non-travel related cases that reported 233,182 contacts.

From the travel-related cases, 888 sequenced genomes were available for comparison to all UK genomic data (see Fig. 1 and Methods for details of case definition and identification,

Supplementary Tables 2 and 3 show the case characteristics). Sequencing of community and hospital cases across the UK was carried out with the aim of providing approximately equal geographical coverage, as much as possible.

Return from European countries accounted for 85.9% (3612/4207) of travel-related cases; 51.2% (2155/4207) had visited one of Greece (21.0%, 882/4207), Croatia (16.3%, 685/4207) or Spain (14.0%,589/4207) (Fig. 1 and Supplementary Table 4). For 284 cases the country of travel was unclear or unknown. Travel restrictions were first eased on 04/07/2020; only 2.9% of travel-related cases identified in this study were recorded before this date. For the countries associated with the highest numbers of imports, the number of cases per day imported from each country along with the timing of travel restriction to that country is shown in Fig. 2. Geographical variations in imported cases across England were apparent, with the greatest number (28.6%, 1205/4207) in Greater London (representing ~15% of the population of England) (Fig. 1 and Supplementary Table 3).

**Contacts per case**. The median number of reported contacts per travel-associated case was 3 (IQR 1–5), with 22% reporting no contacts, while some individuals reported a very large number of contacts, 9% reported more than 10, 3% more than 20, 0.4% more than 50, with a maximum of 172.

Of the imported cases, 2010 were imported from a country with a quarantine requirement at the time of return, whereas 1900 were not required to quarantine on return. For 297 cases quarantine status could not be determined. The number of contacts was higher for cases without a travel restriction (mean = 6.0, median = 3, IQR = 1–7) compared to cases with a travel restriction (mean = 3.0, median = 2, IQR = 1–4).

Using a negative binomial regression model, after adjusting for potential confounding factors of age, sex, date of the test, destination, and ethnicity, travelling from a country requiring quarantine on return was associated with an estimated reduction in the number of contacts of 40% (rate ratio (R.R.) = 0.60, 95% CI = 0.37–0.95; $p = 0.03$). Statistical modelling is fully described in the "Methods" section. Using this model, the estimated marginal mean number of reported contacts (adjusting and averaging over all covariates; age, sex, date of the test, destination, and ethnicity) was 5.85 (95% CI = 3.7–9.3) when no quarantine was required compared to 3.50 (95% CI = 3.0–4.0) when travellers were required to quarantine. To address possible bias from a small number of cases with a large number of contacts, we recorded (top-coded) all cases with more than 10 contacts as corresponding to 10; the estimated rate ratio was slightly attenuated to 0.68 (0.48–0.98; $p = 0.036$) for the number of contacts per case for individuals travelling from a country requiring quarantine on return compared to those not requiring quarantine.

The number of contacts per case varied significantly with age group and over time (Fig. 3 and Supplementary Table 6). The number of contacts per case was greatest in the 16–20 age group who travelled to countries with no requirement for quarantine, with a marginal mean of 9.0 (95% CI = 5.6–14.5) but reduced to 4.7 (95% CI = 3.9–5.7) when quarantine was required—similar to other age-groups.

After adjusting for all other covariates the reported numbers of contacts per imported case was lower in September compared to May, June and July, whether or not a requirement to quarantine was in place. Following this observation of reduced contacts over time among travel-related cases, we compared this to the number of reported contacts over time among the remaining population who did not travel to ensure this was not a general trend due to other COVD-19 measures. Among 105794 cases recorded in our study period that were not associated with travel, 28,564 (27%) were excluded due to poor

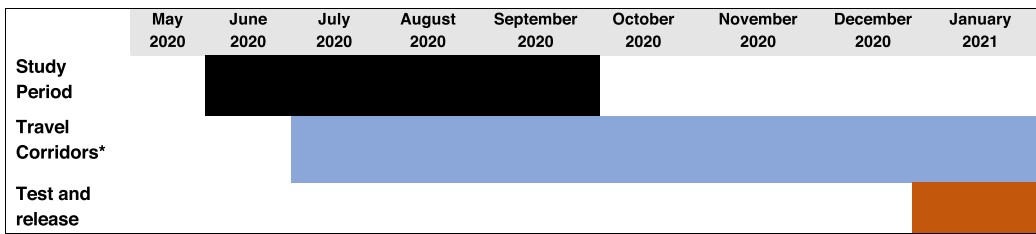

data quality. Among the remaining 77,230 cases we did not find a corresponding decrease over time in the number of contacts, with a mean of 1.6 contacts per case in May/June 2020 and around 2.3 contacts per case for the remaining period.

**Onward transmission and genomic analysis.** We next sought to quantify onward transmission from an imported case using genomics. High-quality sequencing data were available for 827/ 4207 (19.7%) cases (Fig. 1) and demographics of the sequenced

**Fig. 1 Case ascertainment and distribution during the study period. a** Timeline of the study period (27 May 2020 to 13 September 2020) and associated policy changes on travel introduced in England. Travel-related quarantine measures were assigned on a country by country basis from 4 July 2020. Travellers returning from countries that were on the 'closed travel corridors list[58] were required to quarantine for 14 days (*reduced to 10 days on 15/12/20), or from the 15th December 2020, choose to self-isolate for 5 days and then pay for a SARS-CoV-2 diagnostic test (test and release). **b** Flow diagram and map of travel-related cases ascertained from Test and Trace data and subsequent genome availability. Cases were defined as 'highly probable' and 'probable'. 'Highly probable' travel-related cases were defined as individuals who reported international travel as an activity in the two days before symptom onset/testing. On 12/08/2020 the additional facility to report international travel in the 7 days prior to symptom onset/testing became available, and also included in this study and defined as 'probable' travel-related cases. **c** Flow diagram relaying contacts ascertained of cases from Test and Trace data. **d** Countries where importations originated. Countries with less than five importations were excluded for confidentiality reasons. **e** Destinations of imported cases within England. Areas with less than three cases have been excluded. Q/C quality control.

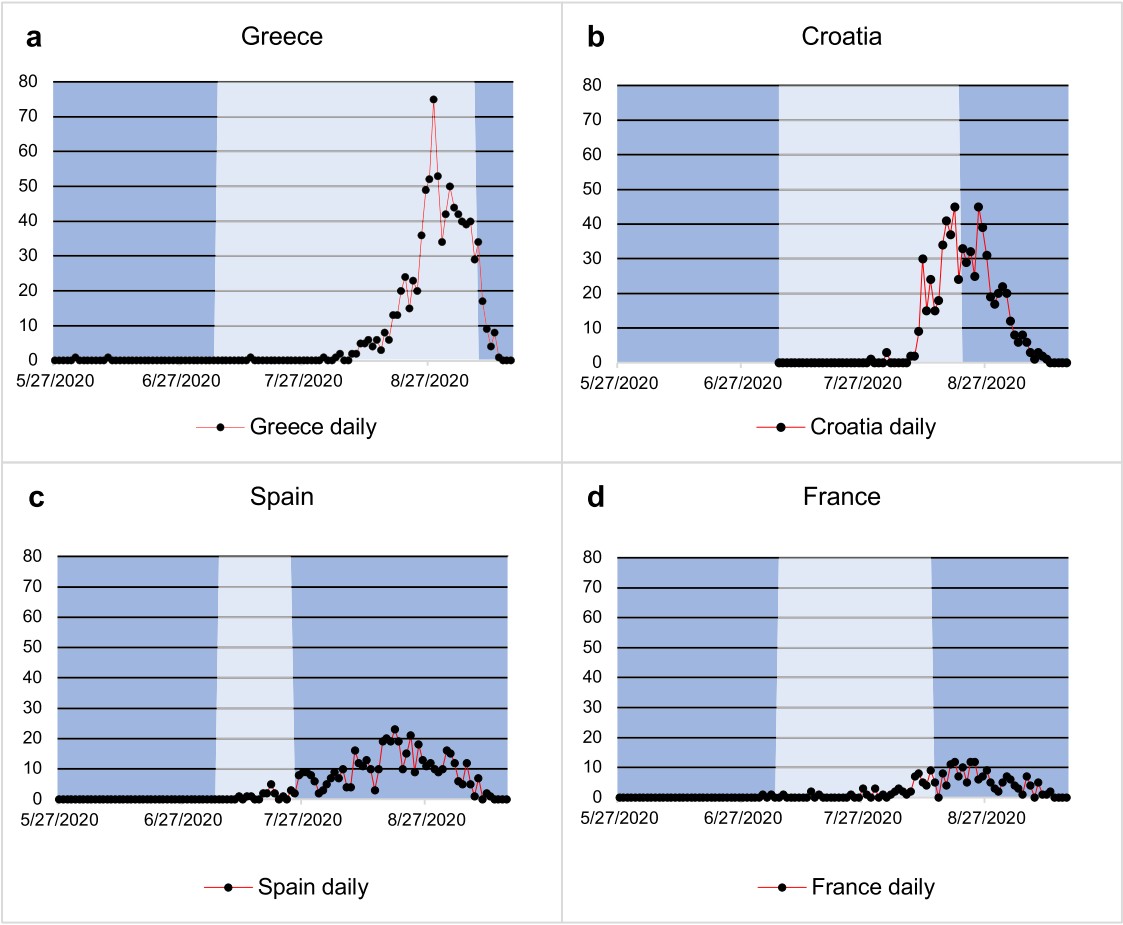

**Fig. 2 Frequency of importations overtime for the top 4 most common countries of travel reported by individuals testing positive for SARS-CoV-2 during the study period. a–d** SARS-CoV-2 case numbers in returning travellers by the four most popular countries of travel reported by cases representing 2379/4207 (56.5%) of known travel-related cases. The light-shaded areas represent the period of time when the countries had an open 'travel-corridor' so did not have mandatory 14-day quarantine on return in place.

cases were broadly similar to the entire travel-related cohort (Supplementary Table 3).

**Important genomes and onward transmission.** To monitor onward spread we identified 186/827 (22.4%) imported cases with SARS-CoV-2 genomes that were sufficiently unique, as defined by their status as extinct or genetically distinct within a sub-lineage (see Methods). Of these, 146/186 isolates had not been sampled in the entire UK dataset in the 4 weeks prior, while a further 40 isolates were more than 3 single nucleotide polymorphisms (SNPs) to their closest matching sequence in the existing UK dataset, both suggesting genuinely new importations of this genotype.

Using an SNP–matrix of imported genomes and associated travel-related metadata, we defined the number of importation events per imported genotype; these ranged from 1 to 39. The majority (119/186; 64%) of genomes were identified only once in imported cases, with 33 (18%) identified twice, 22 (12%) between 3 and 10 times and the remaining 12 (6%) between 11 and the maximum of the observed number of 39.

To compare the effect of the requirement to quarantine on the subsequent spread of likely imported cases, the entire COG-UK dataset was interrogated to identify isolates within 2 SNPs of these distinct imported cases identified up to 4 weeks after the index importation case. There was variation in the number of subsequent (up to 4 weeks later) cases matching each genome

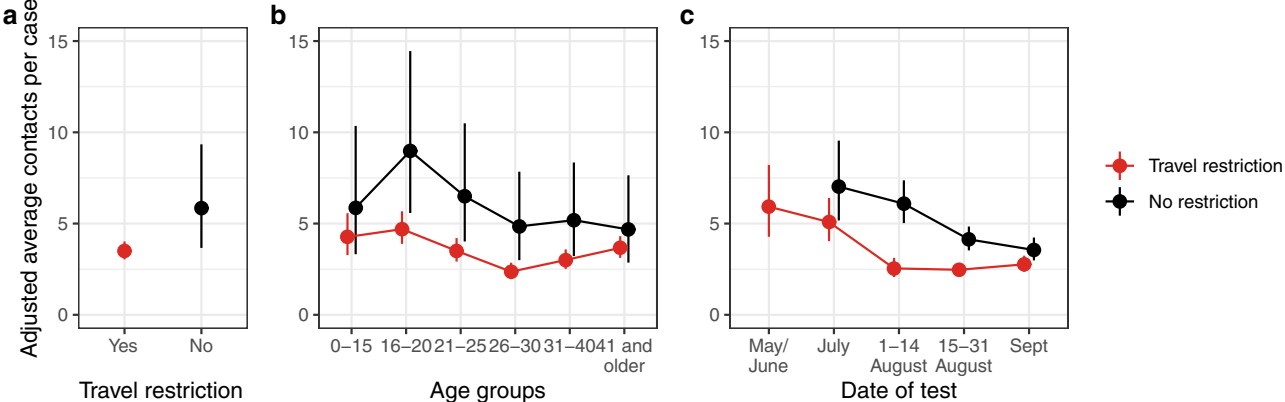

**Fig. 3 The effect of travel restriction (14-day quarantine) on contacts per imported case of SARS-CoV-2.** The estimated marginal mean number of contacts per imported case **a** overall, **b** by age-group and **c** by date of test comparing countries with travel restriction guidance (closed 'travel-corridors') in place and those without (open 'travel-corridors'). All points are estimated marginal means and are provided with 95% confidence intervals.

(median 0; range 0–210, IQR 0–1). The majority of genomes 125/186 = 67% were not linked to any subsequent cases, 17 (9.1%) and 8 (4.3%) were linked to one or two cases, with the remaining 36 (19%) being matched to larger numbers of subsequent cases, with a small number of imported cases corresponding to large numbers of subsequent cases, including 6 that matched to at least 50 later cases.

**Association between travel restriction and onward transmission.** To explore the association between onward transmission and travel restriction we first excluded cases returning before 14 July 2021 to ensure the time periods between index cases with and without a travel restriction overlapped. The proportions of imported genomes matching any subsequent detected case and the number of new cases where at least one is detected in this group are shown in Fig. 4. Overall, 56/168 (33%) of genomes from cases that were genetically unique were detected in at least one subsequent case within the subsequent four weeks. Among genomes identified from a country where quarantine requirement was in place, 25% of (20/81) were detected in at least one subsequent case, compared to 41% (29/71) when cases were imported from a country without a requirement to quarantine (Fig. 4a). The destination country for 16 index cases was unknown.

The number of subsequent cases detected during the 4 weeks since the unique index case increased from a mean of 1.2 new cases when quarantine was required to 11.3 cases where there was no requirement, mainly driven by the fact that all of the nine genomes which went on to match more than 20 subsequent cases had an index case returning from a country without travel restriction (Fig. 4b). However this difference can be explained entirely by the number of importations for each genome; these genomes all had high numbers of independent importations, and genomes with a high number of importations always had an index case returning from countries with no travel restriction in place at the time (Fig. 4c).

To test the statistical significance of observed effects, a series of negative binomial regression models were fitted. In the four weeks following the index case, fewer genomically linked cases were reported when the index case was imported from a country with a requirement to quarantine compared to cases from a country with no requirements (unadjusted R.R. = 0.11, 95% CI = 0.04–0.28), but this effect was entirely explained when the number of imported cases for each genome is included as an 'offset' in the model and adjusting for the date (R.R = 0.83, 95% CI = 0.35–1.92; $p = 0.655$).

There was some evidence that imported cases with higher numbers of contacts for the index case gave rise to more cases in the subsequent month, however, the number of contacts was only known for the index case. This effect is also explained by the number of importations; there is a positive correlation (Spearman's rho = 0.18, $p = 0.018$) between the number of contacts reported by the index case and the number of independent importations of each genome (Supplementary Fig. 1), and in regression models, once we adjust for the number of importations there is no evidence for an association between a number of contacts of the index case and subsequent linked cases. However, it is possible that the number of contacts of other imported cases in each group remains an important factor.

**Genomic identification of a large imported cluster.** In order to demonstrate the utility of genomics in identifying a probable travel-related cluster of SARS-CoV-2 cases in the England de novo, we ran the Polecat Clustering tool (https://cog-uk.github.io/polecat) on 14 September 2020 (including SARS-CoV-2 cases in the COG-UK dataset up to this date). An outlier cluster was observed (Supplementary Fig. 4). This cluster (UK1897) was associated with high diversity with a long stem length compared to samples from the UK, suggesting that this lineage evolved outside the UK. The geographic distribution of this lineage is demonstrated in Supplementary Fig. 5, likely representing multiple importations into the UK. This cluster contained the D614G mutation but no others associated with increased transmission. The root of the cluster was associated with a Swiss phylotype when linked to data in GISAID. During the course of the study period (4 August 2020 to 14 September 2020), there were 304 genomes corresponding to this cluster. These could be linked to 238 individuals, of whom 159 could be linked to a contact-tracing record. Out of 159, 143 had contact-tracing information indicating international travel or not. Out of 143, 72 (50.3%) individuals had recently returned from abroad and were associated with, 10 dispersed European countries (4 individuals had traveled to more than 1 European Country) and most commonly Croatia (35/72, 48.6%) (Supplementary Fig. 6). A further four cases were identified as contacts of individuals who had reported travel to mainland Europe. There is a trend towards an increased proportion of cases that do not report travel over time, and possibly representing dispersion and onwards transmission locally of this lineage (Supplementary Fig. 7).

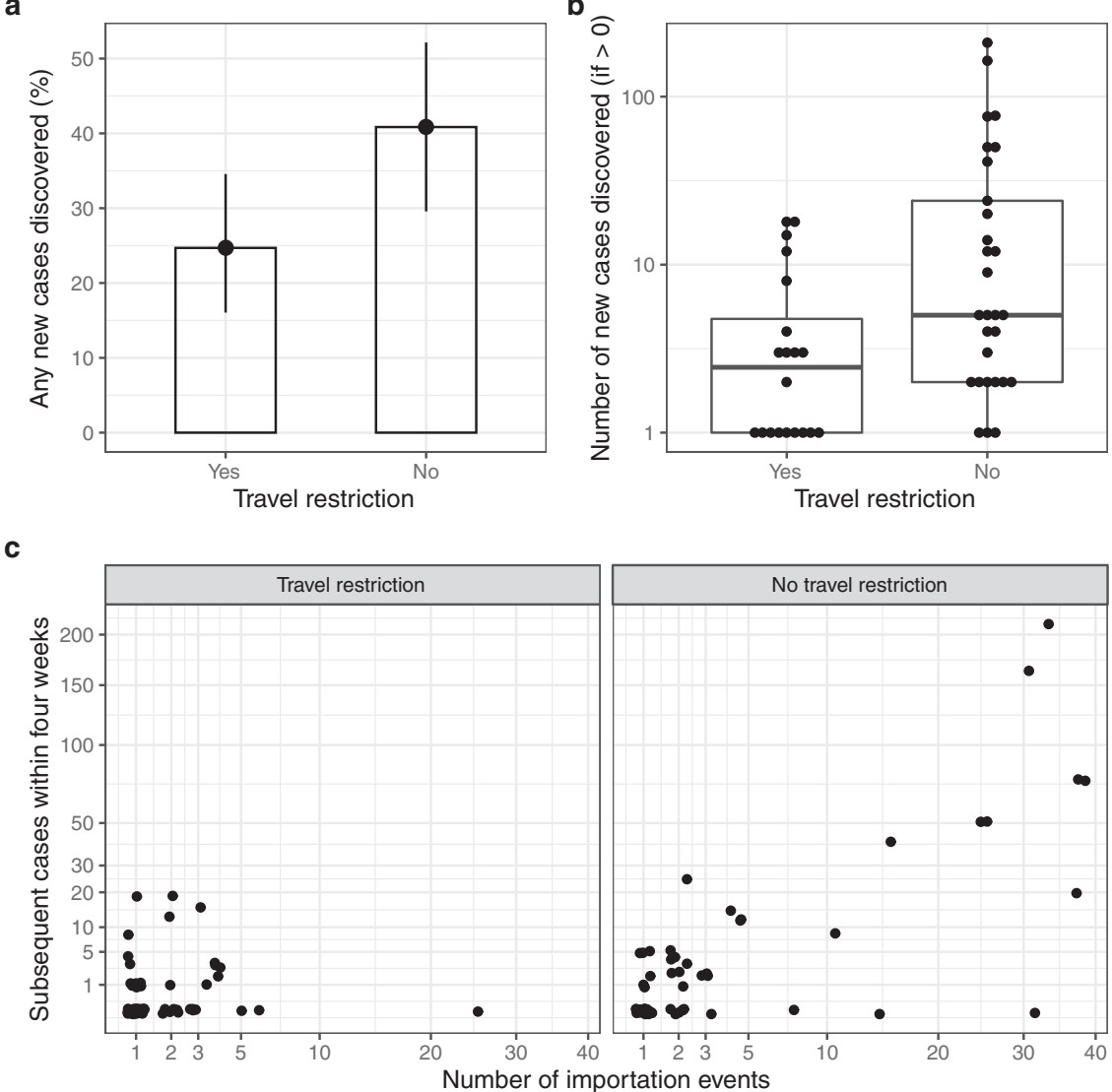

**Fig. 4 The effect of travel restrictions (14-day quarantine) on the subsequent spread of likely imported cases as determined by genomics. a** The proportion of imported cases with any matching genome detected over the four weeks following index test result. Error bars correspond to bootstrapped confidence intervals, $n = 81$ and $71$. **b** The number of genomes matching the index case, with zeros excluded. The midline of the boxplot represents the median value; the lower limit of the box represents the first quartile (25th percentile), and the upper limit of the box represents the third quartile (75th percentile); the whiskers (upper and lower) extend to the largest and smallest value from the box, no further than 1.5*IQR from the box. **a**, **b** Compare countries with travel restriction guidance (closed 'travel-corridors') in place and those without (open 'travel-corridors'). **c** The number of cases and number of importation events for each imported genome, stratified by whether the index case had returned from a country with a travel restriction in place.

**Characteristics of imported genomes.** The 827 imported genomes reflected 238 UK lineages (see Supplementary Materials), of which 214 were seen fewer than 5 times (142 singletons) and 24 were seen 5 or more times (Supplementary Table 8). The most commonly observed were UK5 (152 genomes, 18.4%) and UK1897 (73 genomes, 8.8%). There were 39 global lineages within the genomes. The most commonly observed lineages were B.1.1 (159 genomes, 19.2%) and B.1.177 (128 genomes, 15.5%) (Supplementary Tables 9 and 10). Potentially functionally important mutations were also identified (Supplementary Table 11 and Supplementary Fig. 8): D614G, 824/827 (99.6%) cases; N439K, 65/827 (7.86%) of cases; A222V, 131/827 (15.84%) of cases. ΔH69/V70 was identified in 53 cases associated with lineage B.1.258. We evaluated the introduction of A222V (B.1.177) over time, demonstrating a clear epidemiological link to Spain through contact tracing (Supplementary Fig. 10). By the end of the study period, this variant was introduced from 16 separate countries indicating dispersion across Europe (Supplementary Fig. 11) corroborating findings by Hodcroft et al.[37]. The mutations co-occur, with the proportion of cases represented by these combinations varying over time (Supplementary Fig. 12).

## Discussion

Here, we provide evidence, through the analysis of both contact-tracing data and the use of genomics, that a mandatory 14-day quarantine was associated with fewer contacts for returning travellers with SARS-COV-2, and less onward transmission of imported cases, with the reduced transmission, likely mediated through fewer individual importations of each genome. From 27 May 2020 to 13 September 2020, 85.9% of importations of SARS-COV-2 into England were from European countries with three countries, Greece, Croatia, and Spain, accounting for 51.2% of all imported cases. Along with the requirement to quarantine or not,

age was a significant determinant of onwards contacts, with younger age groups reporting more contacts, but the effect of a requirement to quarantine on the number of contacts was observed across all age groups. We have shown that after a period of national lockdown, systematic monitoring of imported genomes can identify sequences that are sufficiently unique and provide utility for monitoring onwards transmission.

Whilst the study period covers nearly 5 months, the importations were concentrated after the implementation of travel corridors; prior to this date travel was not advised[38]. The peaks for imports for each country occur at different times and with different epidemic curves, likely affected by both patterns of travel as well as the prevalence of disease and local regulations within each country. For the most common destinations, barring Spain, imported cases appear to reduce after the closing of a travel corridor and subsequent requirement to quarantine. The majority of importations from Greece came at the end of August and continued into September; there was no requirement to quarantine for travellers returning from Greece during this time period and it was the source of the greatest imported SARS-CoV-2 cases during the study period. This highlights the need for active surveillance of imported cases of SARS-CoV-2 for the introduction of requirements to quarantine in a timely manner. London accounts for 15.4% of the population in England[39] and observed 11.4% (12011/105794) SARS-CoV-2 infections during the study period; the region however accounted for 28.6% of imported SARS-CoV-2 cases, possibly reflecting a younger and more diverse demographic with cultural/family links abroad, and with a concentration of international businesses and airports. The reported effective reproduction number (Rε) in London had a minimum lower-bound value of 0.6 and an upper-bound value of 1.3 during the study period which was comparably similar to the respective values of 0.7 and 1.2 observed for England[40]. This potentially indicates imports are unlikely to have had a substantial impact on onward infection rates in this region.

The number of onwards contacts were significantly lower when the traveller was required to quarantine highlighting the effectiveness of this policy. Age was also a significant determinant of onwards contacts, with the 16–20-year-old age group representing the greatest number of travel-related cases and the greatest number of onwards contacts per case. This identifies an opportunity to direct public health awareness campaigns to younger travellers, with the intention to promote behaviours that will reduce the risk of SARS-CoV-2 acquisition and enhance compliance with a quarantine on return.

We observed a reduction in the number of contacts per case over the Summer 2020 period that appeared specific to the travel-relatedSARS-CoV-2 cases as it was not replicated in the non-travel related cases. There was also no apparent change in successful contact tracing to explain this difference (Supplementary Fig. 13). We speculate this observation may be related to a change in traveller behaviour (e.g. due to rising cases in the England or destination countries) or changes in types of traveller (e.g. travellers visiting family versus those for occupational reasons or those with dependents versus those without). Understanding these temporal changes in traveller behaviour requires formal investigation with more detailed epidemiological information available and replication in other countries. More broadly, we do find non-travellers reported fewer contacts per case when compared to travellers which may reflect a more sociable cohort, and therefore one that may benefit from targeted public health messaging to reduce transmission risk.

The use of genomic sequencing allowed the identification of a cohort of unique genomes that could be monitored for cluster growth. The cluster size for genomes that were imported from a country without a requirement to quarantine on return was significantly higher than those related to countries with mandatory quarantine in place providing further assurance on the effectiveness of quarantine policy on reducing travel-relatedSARS-CoV-2 cases. This finding was explained by several large clusters, all of which came from countries with no quarantine requirement at the time and with high numbers of individual importation events. With the number of importation events per cluster taken into account, we did not observe an effect of quarantine on subsequent cluster size suggesting the largest effect of travel-related quarantine is through a net reduction in travel-related importations.

The Polecat Clustering Tool (https://cog-uk.github.io/polecat) highlighted a large cluster that developed largely through travel to Croatia. This analysis shows that programmatic analysis of genomics data can identify putative importation clusters. Integration with contact-tracing information was vital for the true picture of the sources of introduction and the subsequent spread, due to the bias of SARS-CoV-2 sequencing globally[41]. In this instance, an introduced lineage was associated with widespread dispersal and onward transmission during a period when England had limited social distancing measures[42,43]. The lineage, B.1.160, associated with this cluster is not associated with increased transmissibility but this method for the detection of expanding imported clusters could be useful for the investigation of newly introduced variants of concern.

Our study has several important limitations. The COG-UK dataset has a limited sequencing coverage across England meaning cluster sizes detected will under-estimate absolute numbers and there is a possibility of unsampled transmission chains despite our use of four SNPs to identify divergent clusters (Supplementary Fig. 14). The earliest reliable data available to identify if individuals were required to quarantine on returning to England from travel abroad was the date of case sampling. Our study evaluates a period of time following a national lockdown with highly restricted movement across borders which likely exaggerated the diversity of imported genomes compared to lineages circulating in England. Additionally, the quarantine guidance at this time was of 14 days; shorter periods may be as efficacious and/or when combined with testing[44]. Outcomes such as travel and the number of contacts are self-reported; reporting bias is mitigated through mandatory completion of a passenger locator form to assist identification of returning travellers, while travel-related cases are seen as higher risk and therefore referred to local public health agencies for targeted contact tracing. For genomic analysis, only the destination country and number of contacts for the index case in each cluster are used, irrespective of the number of imports. Finally, there will be an artificial reduction in cases at the end of the study period when accounting for the case incubation period, testing and report, with data provided 3 days after study close.

Overall, we present an integrated epidemiological and genomic evaluation of the largest dataset of confirmed SARS-CoV-2 imported cases into England (or any other country) to our knowledge. This study provides evidence for the effectiveness of 14-day quarantine in reducing contacts, and reducing, but not entirely preventing onward transmission of imported cases, through reducing the number of importations. Our data highlights the possibility of targeted public health campaigns to reduce SARS-CoV-2 importations and onwards transmission. In conclusion, this study demonstrates how routine genomic epidemiology of travel-related cases could be used to monitor SARS-CoV-2 import cases to enable rapid refinement of travel policies.

## Methods

**Ethics**. The COG-UK study protocol was approved by the Public Health England Research Ethics Governance Group (reference: R&D NR0195). Public Health

England affiliated authors had access to identifiable Cambridgeshire community case data. This data was processed under Regulation 3 of The Health Service (Control of Patient Information) Regulations 2002—permitting the processing of confidential patient information for communicable disease and other risks to public health and as such, individual patient consent is not required. Other authors only had access to anonymised or summarised data.

**Contact tracing and case identification**. Contact-tracing data was obtained from Test and Trace (T&T). All cases and contacts had a field for demographic data, but this was not always reported (Supplementary Tables 3 and 4). 'Highly probable' travel-related cases were defined as individuals who reported international travel as an activity in the two days before symptom onset/testing. On 12/08/2020 the additional facility to report international travel in the seven days prior to symptom onset/testing became available, and also included in this study and defined as 'probable' travel-related cases.

Cases were asked to provide details of all contacts for activities in the 2 days prior to symptom-onset/testing (whichever is earliest) up to the time of completing the system in which contacts were gathered. Though there can be a discrepancy in the time taken for individuals to complete the contact tracing system, this is not expected to result in a material change in contact numbers as they are expected to be self-isolating after symptom-onset or testing positive. If any contacts become cases they would then also be included in T&T data as a case separately, but if they did not report direct travel themselves, then they would not meet the definition for a travel-associated case.

A contact is defined as an individual who a case has had face-to-face contact with (less than 1 metre away), spent more than 15 min within 2 m of, travelling in a car or other small vehicle with, or sat close to them on a plane

**Case identification from T&T data**. Data included free-text destination city or country. A free text country and city search with a custom python script on travel-related T&T were used to identify destination country. Results and remaining entries were manually checked and corrected (see Supplementary materials for more details).

**Requirement for quarantine**. Persons returning from countries where travel-related quarantine was not mandated (except for exemptions e.g. specific employment) were required to quarantine for 14 days at home. The package of measures used to help enforce this requirement included the need to complete a Passenger Locator Form prior to arrival in England, spot checks by the PHE Isolation Assurance Service, and referral to the police through the Border Force Criminal Justice Unit who may issue fixed penalty notices.

**Clinical samples, genome sequencing and quality control**. Clinical samples were collected passively as part of national SARS-CoV-2 testing. This included both community testing through lighthouse labs (satellite SARS-CoV-2 testing laboratories) and testing through hospital diagnostic labs. Samples were sequenced at one of seventeen COG-UK sequencing sites (Fig. 1). The samples were prepared for sequencing using either the ARTIC[45] or veSeq[46] protocols and were sequenced using Illumina or Oxford Nanopore platforms. All samples were uploaded to and processed through COVID-CLIMB pipelines[47,48]. Genomes were aligned to the Wuhan Hu-1 reference genome (Genbank accession code: MN908947.3). Genomes that contained more than 10% missing data were excluded from further analysis to ensure high-quality phylogenetic analysis.

**Lineages and minor variants**. Global and UK Lineages[49] were assigned to each genome using Pangolin (https://github.com/cov-lineages/pangolin) with analysis performed on COVID-CLIMB[48]. Minor variants were pre-defined within the COG-UK database using type_variants (https://github.com/cov-ert/type_variants).

**Identification of extinct and unique genomes**. The 827 high-quality travel-related genomes were compared to the COG-UK dataset on 16/10/2020. Genomes were only compared to other genomes with the same UK lineage assigned by COG-UK since we assume that no relatedness relevant to transmission exists between genomes of different UK lineages. A unique genome in the community was deemed to be one that was known to be from a travel-related case and that was either: a UK lineage that had not been sampled in the previous 4 weeks in the UK or was more than 3 SNPs distance to the closest relative in the COG-UK dataset.

Within the same UK lineage, we then identified those genomes sampled within 4 weeks prior to the genome of interest. We determined the SNP distance between the sequence of interest and these genomes. Unique genomes were compared to sequences that were generated in the COG-UK dataset within 2 and 4 weeks after their sampling date, to identify samples with the same UK lineage and within 2 SNPs. These would represent onward transmission or further introductions of similar genomes. The analysis was run with an in-house custom Python script developed by US and RM. Further detail in Supplementary materials.

**Identification of multiple introductions of a unique genome**. We combined the available travel-related epidemiological data and genomic data to identify the

number of importations representing the travel-related clusters generated above. We used SNP-dist (version0.7.0) (https://github.com/tseemann/snp-dists) to identify the SNP distances between an alignment 827 high-quality imported genomes. These genomes were aligned with MAFFT (version7.471)[50], outside of CLIMB-COVID pipelines, with minor differences in SNP differences to the entire COG-UK alignment expected. We then identified imported genomes that were within 2 SNPs of the 186 unique imported genomes in the 4 weeks subsequent to the unique imported case being sampled. This represented the number of importations of that genome in the 4-week period of interest corresponding to each unique travel-related cluster identified in the analysis above.

**Identification of a travel-related SARS-CoV-2 cluster**. We used the Polecat clustering tool (https://cog-uk.github.io/polecat) to systematically identify outliers in COG-UK genomic dataset and link to contact-tracing data.

**Statistical analysis**. Statistical models for the number of contacts per case and the number of onward cases per imported genotype were estimated using the glmmTMB package (version 1.0.1)[51] with marginal means and effects calculated using the emmeans package (1.5.2-1)[52] for R (version 4.02)[53]. Figures were generated using R (version 4.0.2) and Microsoft Excel (version 1908).

The number of contacts per case was modelled using negative binomial regression, to estimate the effect of travel-related quarantine, and whether this varied by age group, sex of the index case and calendar date. Travel destination and ethnic group were included as random effects. Negative binomial regression models were also used for the number of onward cases per imported genome, using calendar date as a covariate and the natural log of the number of importation events for each genome as an offset variable. Model validation was performed by simulation from estimated models and comparing the distribution of observed and modelled outcomes. For the contacts modelling the initial negative binomial regression model did not reflect the number of large outliers seen in the original data, therefore additional models were estimated using data with these observations dropped or top-coded to check that the reported estimates of effects were not driven by these observations. For the genomic data, the simulated and observed distributions were closely aligned.

**Reporting summary**. Further information on research design is available in the Nature Research Reporting Summary linked to this article.

## Data availability

Assembled/consensus genomes are available from GISAID[54] subject to minimum quality control criteria. Raw reads are available from European Nucleotide Archive (ENA)[55] under accession PRJEB37886. ENA accession codes for travel-related genomes used in this study are available in supplementary materials (Supplementary Data 1) and available from GitHub at https://github.com/COG-UK/travel-quarantine/[56]. The Genbank accession code for the reference genome uses is MN908947.3. All genomes, phylogenetic trees, basic metadata are available from the COG-UK consortium website (https://www.cogconsortium.uk/data). Extensive aggregated metadata has been made available in supplementary files. Genomes accessed through GISAID used in this study are provided in the Supplementary Information file entitled 'GISAID acknowledgement table'. For confidentiality reasons[57], extended metadata are under restricted access; requests for access should be directed to corresponding authors and specifically for Public Health England data, to the Public Health England office of data release (https://www.gov.uk/government/publications/accessing-public-health-england-data/about-the-phe-odr-and-accessing-data) with an estimated 60 working days turnaround time.

## Code availability

Custom code used in this analysis is available at https://github.com/COG-UK/travel-quarantine/[56]. Please direct further queries to the corresponding authors.

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

## Acknowledgements

We thank members of the COVID-19 Genomics Consortium UK and Test and Trace contact tracers for their contributions to generating the data used in this study. We thank the Sanger Covid Team for assisting with Samples and Logistics. We thank Sarah Mitchell from the Department of Plant Sciences, University of Cambridge, for direction on statistical analysis. D.A. is a Clinical PhD Fellow and gratefully supported by the Wellcome Trust (Grant no. 222903/Z/21/Z). E.M.H. is supported by a UK Research and Innovation (UKRI) Fellowship: MR/S00291X/1. A.J.P., T.L.V. and G.M.S. gratefully acknowledge the support of

the Biotechnology and Biological Sciences Research Council (BBSRC); their research was funded by the BBSRC Institute Strategic Programme Microbes in the Food Chain BB/R012504/1 and its constituent project BBS/E/F/000PR10352, also Quadram Institute Bioscience BBSRC funded Core Capability Grant (project number BB/CCG1860/1). The COVID-19 Genomics UK (COG-UK) Consortium is supported by funding from the Medical Research Council (MRC) part of UK Research & Innovation (UKRI), the National Institute of Health Research (NIHR) and Genome Research Limited, operating as the Wellcome Sanger Institute. The funders had no role in study design, data collection and analysis, decision to publish, or preparation of the manuscript. For the purpose of open access, the author has applied a CC-BY public copyright licence to any Author Accepted Manuscript version arising from this submission.

## Author contributions

Conceptualisation: M.C., E.M.H., S.J.P. and D.A.; Methodology: D.A., U.S., R.M., E.V., M.C. and E.M.H.; Supervision: M.C., E.M.H. and S.J.P.; Writing—original draft preparation: D.A., A.J.P. and G.M.S.; Writing—review and editing: All authors; Investigation: D.A., U.S., A.J.P., R.M., N.E., T.L.V. and N.M.T.; Formal analysis: D.A., U.S., R.M. and G.M.S.; Data curation: D.A., S.P., N.G., E.G., G.J.H., C.C., C.T., S.L. and A.H.; Funding acquisition: S.J.P. and E.M.H.; Resources and validation: G.J.H., C.C., C.T., S.L., A.H., D.A., A.J.P. and U.S.; Resources: COG-UK consortium.

## Competing interests

The authors declare no competing interests.

## Additional information

## The COVID-19 Genomics UK (COG-UK) Consortium

Cherian Koshy[11], Amy Ash[11], Emma Wise[12], Nathan Moore[12], Matilde Mori[12], Nick Cortes[12], Jessica Lynch[12], Stephen Kidd[12], Derek J. Fairley[13], Tanya Curran[13], James P. McKenna[13], Helen Adams[14], Christophe Fraser[15], Tanya Golubchik[15], David Bonsall[15], Mohammed O. Hassan-Ibrahim[16], Cassandra S. Malone[16], Benjamin J. Cogger[16], Michelle Wantoch[17], Nicola Reynolds[17], Ben Warne[3], Joshua Maksimovic[18], Karla Spellman[18], Kathryn McCluggage[18], Michaela John[18], Robert Beer[18], Safiah Afifi[18], Sian Morgan[18], Angela Marchbank[19], Anna Price[19], Christine Kitchen[19], Huw Gulliver[19], Ian Merrick[19], Joel Southgate[19], Martyn Guest[19], Robert Munn[19], Trudy Workman[19], Thomas R. Connor[19], William Fuller[19], Catherine Bresner[19], Luke B. Snell[20], Amita Patel[20], Themoula Charalampous[21], Gaia Nebbia[21], Rahul Batra[21], Jonathan Edgeworth[21], Samuel C. Robson[22], Angela H. Beckett[22], David M. Aanensen[23], Anthony P. Underwood[23], Corin A. Yeats[23], Khalil Abudahab[23], Ben E. W. Taylor[23], Mirko Menegazzo[23], Gemma Clark[24], Wendy Smith[24], Manjinder Khakh[24], Vicki M. Fleming[24], Michelle M. Lister[24], Hannah C. Howson-Wells[24], Louise Berry[24], Tim Boswell[24], Amelia Joseph[24], Iona Willingham[24], Carl Jones[24], Christopher Holmes[25], Paul Bird[25], Thomas Helmer[25], Karlie Fallon[25], Julian Tang[25], Veena Raviprakash[26], Sharon Campbell[26], Nicola Sheriff[26], Victoria Blakey[26], Lesley-Anne Williams[26], Matthew W. Loose[27], Nadine Holmes[27], Christopher Moore[27], Matthew Carlile[27], Victoria Wright[27], Fei Sang[27], Johnny Debebe[27], Francesc Coll[28], Adrian W. Signell[29], Gilberto Betancor[29], Harry D. Wilson[29], Sahar Eldirdiri[30], Anita Kenyon[30], Thomas Davis[30], Oliver G. Pybus[31], Louis du Plessis[31], Alex E. Zarebski[31], Jayna Raghwani[31], Moritz U. G. Kraemer[31], Sarah Francois[31], Stephen W. Attwood[31], Tetyana I. Vasylyeva[31], Marina Escalera Zamudio[31], Bernardo Gutierrez[31], M. Estee Torok[32], William L. Hamilton[32], Ian G. Goodfellow[33], Grant Hall[33], Aminu S. Jahun[33], Yasmin Chaudhry[33], Myra Hosmillo[33], Malte L. Pinckert[33], Iliana Georgana[33], Samuel Moses[34], Hannah Lowe[34], Luke Bedford[35], Jonathan Moore[36], Susanne Stonehouse[36], Chloe L. Fisher[37], Ali R. Awan[37], John BoYes[38], Judith Breuer[39], Kathryn A. Harris[39], Julianne R. Brown[39], Divya Shah[39], Laura Atkinson[39], Jack C. D. Lee[39],

Nathaniel Storey[39], Flavia Flaviani[40], Adela Alcolea-Medina[41], Rebecca Williams[42], Gabrielle Vernet[42], Michael R. Chapman[43], Lisa J. Levett[44], Judith Heaney[44], Wendy Chatterton[44], Monika Pusok[44], Li Xu-McCrae[45], Darren L. Smith[46], Matthew Bashton[46], Gregory R. Young[46], Alison Holmes[47], Paul A. Randell[47], Alison Cox[47], Pinglawathee Madona[47], Frances Bolt[47], James Price[47], Siddharth Mookerjee[47], Manon Ragonnet-Cronin[6], Fabricia F. Nascimento[6], David Jorgensen[6], Igor Siveroni[6], Rob Johnson[6], Olivia Boyd[6], Lily Geidelberg[6], Erik M. Volz[6], Aileen Rowan[6], Graham P. Taylor[6], Katherine L. Smollett[48], Nicholas J. Loman[49], Joshua Quick[49], Claire McMurray[49], Joanne Stockton[49], Sam Nicholls[49], Will Rowe[49], Radoslaw Poplawski[49], Alan McNally[49], Rocio T. Martinez Nunez[50], Jenifer Mason[51], Trevor I. Robinson[51], Elaine O'Toole[51], Joanne Watts[51], Cassie Breen[51], Angela Cowell[51], Graciela Sluga[52], Nicholas W. Machin[53], Shazaad S. Y. Ahmad[53], Ryan P. George[53], Fenella Halstead[54], Venkat Sivaprakasam[54], Wendy Hogsden[54], Chris J. Illingworth[55], Chris Jackson[55], Emma C. Thomson[56], James G. Shepherd[56], Patawee Asamaphan[56], Marc O. Niebel[56], Kathy K. Li[56], Rajiv N. Shah[56], Natasha G. Jesudason[56], Lily Tong[56], Alice Broos[56], Daniel Mair[56], Jenna Nichols[56], Stephen N. Carmichael[56], Kyriaki Nomikou[56], Elihu Aranday-Cortes[56], Natasha Johnson[56], Igor Starinskij[56], Ana da Silva Filipe[56], David L. Robertson[56], Richard J. Orton[56], Joseph Hughes[56], Sreenu Vattipally[56], Joshua B. Singer[56], Seema Nickbakhsh[56], Antony D. Hale[57], Louissa R. Macfarlane-Smith[57], Katherine L. Harper[57], Holli Carden[57], Yusri Taha[58], Brendan A. I. Payne[58], Shirelle Burton-Fanning[58], Sheila Waugh[58], Jennifer Collins[58], Gary Eltringham[58], Steven Rushton[59], Sarah O'Brien[59], Amanda Bradley[60], Alasdair Maclean[60], Guy Mollett[60], Rachel Blacow[60], Kate E. Templeton[61], Martin P. McHugh[61], Rebecca Dewar[61], Elizabeth Wastenge[61], Samir Dervisevic[62], Rachael Stanley[62], Emma J. Meader[62], Lindsay Coupland[62], Louise Smith[63], Clive Graham[64], Edward Barton[64], Debra Padgett[64], Garren Scott[64], Emma Swindells[65], Jane Greenaway[65], Andrew Nelson[66], Clare M. McCann[66], Wen C. Yew[66], Monique Andersson[67], Timothy Peto[67], Anita Justice[67], David Eyre[67], Derrick Crook[67], Tim J. Sloan[68], Nichola Duckworth[68], Sarah Walsh[68], Anoop J. Chauhan[69], Sharon Glaysher[69], Kelly Bicknell[69], Sarah Wyllie[69], Scott Elliott[69], Allyson Lloyd[69], Robert Impey[69], Nick Levene[70], Lynn Monaghan[70], Declan T. Bradley[71], Tim Wyatt[71], Elias Allara[72], Clare Pearson[72], Husam Osman[72], Andrew Bosworth[72], Esther Robinson[72], Peter Muir[72], Ian B. Vipond[72], Richard Hopes[72], Hannah M. Pymont[72], Stephanie Hutchings[72], Martin D. Curran[73], Surendra Parmar[73], Angie Lackenby[2], Tamyo Mbisa[2], Steven Platt[2], Shahjahan Miah[2], David Bibby[2], Carmen Manso[2], Jonathan Hubb[2], Meera Chand[2,105], Gavin Dabrera[2], Mary Ramsay[2], Daniel Bradshaw[2], Alicia Thornton[2], Richard Myers[2], Ulf Schaefer[2,104], Natalie Groves[2], Eileen Gallagher[2], David Lee[2], David Williams[2], Nicholas Ellaby[2], Ian Harrison[2], Hassan Hartman[2], Nikos Manesis[2], Vineet Patel[2], Chloe Bishop[2], Vicki Chalker[2], Juan Ledesma[2], Katherine A. Twohig[2], Matthew T. G. Holden[74], Sharif Shaaban[74], Alec Birchley[75], Alexander Adams[75], Alisha Davies[75], Amy Gaskin[75], Amy Plimmer[75], Bree Gatica-Wilcox[75], Caoimhe McKerr[75], Catherine Moore[75], Chris Williams[75], David Heyburn[75], Elen De Lacy[75], Ember Hilvers[75], Fatima Downing[75], Giri Shankar[75], Hannah Jones[75], Hibo Asad[75], Jason Coombes[75], Joanne Watkins[75], Johnathan M. Evans[75], Laia Fina[75], Laura Gifford[75], Lauren Gilbert[75], Lee Graham[75], Malorie Perry[75], Mari Morgan[75], Matthew Bull[75], Michelle Cronin[75], Nicole Pacchiarini[75], Noel Craine[75], Rachel Jones[75], Robin Howe[75], Sally Corden[75], Sara Rey[75], Sara Kumziene-SummerhaYes[75], Sarah Taylor[75], Simon Cottrell[75], Sophie Jones[75], Sue Edwards[75], Justin O'Grady[5], Andrew J. Page[5,104], Alison E. Mather[5], David J. Baker[5], Steven Rudder[5], Alp Aydin[5], Gemma L. Kay[5], Alexander J. Trotter[5], Nabil-Fareed Alikhan[5], Leonardo de Oliveira Martins[5], Thanh Le-Viet[5], Lizzie Meadows[5], Anna Casey[76], Liz Ratcliffe[76], David A. Simpson[77], Zoltan Molnar[77], Thomas Thompson[77], Erwan Acheson[77], Jane A. H. Masoli[78], Bridget A. Knight[78], Sian Ellard[78], Cressida Auckland[78], Christopher R. Jones[78], Tabitha W. Mahungu[79], Dianne Irish-Tavares[79], Tanzina Haque[79], Jennifer Hart[79], Eric Witele[79], Melisa L. Fenton[80], Ashok Dadrah[80],

Amanda Symmonds[80], Tranprit Saluja[80], Yann Bourgeois[81], Garry P. Scarlett[81], Katie F. Loveson[82], Salman Goudarzi[82], Christopher Fearn[82], Kate Cook[82], Hannah Dent[82], Hannah Paul[82], David G. Partridge[83], Mohammad Raza[83], Cariad Evans[83], Kate Johnson[83], Steven Liggett[84], Paul Baker[84], Stephen Bonner[84], Sarah Essex[84], Ronan A. Lyons[85], Kordo Saeed[86], Adhyana I. K. Mahanama[86], Buddhini Samaraweera[86], Siona Silveira[86], Emanuela Pelosi[86], Eleri Wilson-Davies[86], Rachel J. Williams[87], Mark Kristiansen[87], Sunando Roy[87], Charlotte A. Williams[87], Marius Cotic[87], Nadua Bayzid[87], Adam P. Westhorpe[87], John A. Hartley[87], Riaz Jannoo[87], Helen L. Lowe[87], Angeliki Karamani[87], Leah Ensell[87], Jacqui A. Prieto[88], Sarah Jeremiah[88], Dimitris Grammatopoulos[89], Sarojini Pandey[89], Lisa Berry[89], Katie Jones[89], Alex Richter[90], Andrew Beggs[90], Angus Best[91], Benita Percival[91], Jeremy Mirza[91], Oliver Megram[91], Megan Mayhew[91], Liam Crawford[91], Fiona Ashcroft[91], Emma Moles-Garcia[91], Nicola Cumley[91], Colin P. Smith[92], Giselda Bucca[92], Andrew R. Hesketh[92], Beth Blane[93], Sophia T. Girgis[93], Danielle Leek[93], Sushmita Sridhar[93], Sally Forrest[93], Claire Cormie[93], Harmeet K. Gill[93], Joana Dias[93], Ellen E. Higginson[93], Mailis Maes[93], Jamie Young[93], Leanne M. Kermack[93], Ravi K. Gupta[93], Catherine Ludden[93], Sharon J. Peacock [93], Sophie Palmer[93], Carol M. Churcher[93], Nazreen F. Hadjirin[93], Alessandro M. Carabelli[93], Ellena Brooks[93], Kim S. Smith[93], Katerina Galai[93], Georgina M. McManus[93], Chris Ruis[93], Rose K. Davidson[94], Andrew Rambaut[95], Thomas Williams[95], Carlos E. Balcazar[95], Michael D. Gallagher[95], Áine O'Toole[95], Stefan Rooke[95], Verity Hill[95], Kathleen A. Williamson[95], Thomas D. Stanton[95], Stephen L. Michell[96], Claire M. Bewshea[96], Ben Temperton[96], Michelle L. Michelsen[96], Joanna Warwick-Dugdale[96], Robin Manley[96], Audrey Farbos[96], James W. Harrison[96], Christine M. Sambles[96], David J. Studholme[96], Aaron R. Jeffries[96], Alistair C. Darby[97], Julian A. Hiscox[97], Steve Paterson[97], Miren Iturriza-Gomara[97], Kathryn A. Jackson[97], Anita O. Lucaci[97], Edith E. Vamos[97], Margaret Hughes[97], Lucille Rainbow[97], Richard Eccles[97], Charlotte Nelson[97], Mark Whitehead[97], Lance Turtle[97], Sam T. Haldenby[97], Richard Gregory[97], Matthew Gemmell[97], Claudia Wierzbicki[97], Hermione J. Webster[97], Thushan I. de Silva[98], Nikki Smith[98], Adrienn Angyal[98], Benjamin B. Lindsey[98], Danielle C. Groves[98], Luke R. Green[98], Dennis Wang[98], Timothy M. Freeman[98], Matthew D. Parker[98], Alexander J. Keeley[98], Paul J. Parsons[98], Rachel M. Tucker[98], Rebecca Brown[98], Matthew Wyles[98], Max Whiteley[98], Peijun Zhang[98], Marta Gallis[98], Stavroula F. Louka[98], Chrystala Constantinidou[99], Meera Unnikrishnan[99], Sascha Ott[99], Jeffrey K. J. Cheng[99], Hannah E. Bridgewater[99], Lucy R. Frost[99], Grace Taylor-Joyce[99], Richard Stark[99], Laura Baxter[99], Mohammad T. Alam[99], Paul E. Brown[99], Dinesh Aggarwal [93], Alberto C. Cerda[100], Tammy V. Merrill[100], Rebekah E. Wilson[100], Patrick C. McClure[101], Joseph G. Chappell[101], Theocharis Tsoleridis[101], Jonathan Ball[101], David Buck[102], John A. Todd[102], Angie Green[102], Amy Trebes[102], George MacIntyre-Cockett[102], Mariateresa de Cesare[102], Alex Alderton[4], Roberto Amato[4], Cristina V. Ariani[4], Mathew A. Beale[4], Charlotte Beaver[4], Katherine L. Bellis[4], Emma Betteridge[4], James Bonfield[4], John Danesh[4], Matthew J. Dorman[4], Eleanor Drury[4], Ben W. Farr[4], Luke Foulser[4], Sonia Goncalves[4], Scott Goodwin[4], Marina Gourtovaia[4], Ewan M. Harrison[4], David K. Jackson[4], Dorota Jamrozy[4], Ian Johnston[4], Leanne Kane[4], Sally Kay[4], Jon-Paul Keatley[4], Dominic Kwiatkowski[4], Cordelia F. Langford[4], Mara Lawniczak[4], Laura Letchford[4], Rich Livett[4], Stephanie Lo[4], Inigo Martincorena[4], Samantha McGuigan[4], Rachel Nelson[4], Steve Palmer[4], Naomi R. Park[4], Minal Patel[4], Liam Prestwood[4], Christoph Puethe[4], Michael A. Quail[4], Shavanthi Rajatileka[4], Carol Scott[4], Lesley Shirley[4], John Sillitoe[4], Michael H. Spencer Chapman[4], Scott A. J. Thurston[4], Gerry Tonkin-Hill[4], Danni Weldon[4], Diana Rajan[4], Iraad F. Bronner[4], Louise Aigrain[4], Nicholas M. Redshaw[4], Stefanie V. Lensing[4], Robert Davies[4], Andrew Whitwham[4], Jennifer Liddle[4], Kevin Lewis[4], Jaime M. Tovar-Corona[4], Steven Leonard[4], Jillian Durham[4], Andrew R. Bassett[4], Shane McCarthy[4], Robin J. Moll[4], Keith James[4], Karen Oliver[4], Alex Makunin[4], Jeff Barrett[4] & Rory N. Gunson[103]

[11]Barking, Havering and Redbridge University Hospitals NHS Trust, Romford, UK. [12]Basingstoke Hospital, Basingstoke, UK. [13]Belfast Health & Social Care Trust, Belfast, UK. [14]Betsi Cadwaladr University Health Board, Bangor, UK. [15]Big Data Institute, Nuffield Department of Medicine, University of Oxford, Oxford, UK. [16]Brighton and Sussex University Hospitals NHS Trust, Brighton, UK. [17]Cambridge Stem Cell Institute, University of Cambridge, Cambridge, UK. [18]Cardiff and Vale University Health Board, Cardiff, UK. [19]Cardiff University, Cardiff, UK. [20]Centre for Clinical Infection & Diagnostics Research, St. Thomas' Hospital and Kings College London, London, UK. [21]Centre for Clinical Infection and Diagnostics Research, Department of Infectious Diseases, Guy's and St Thomas' NHS Foundation Trust, London, UK. [22]Centre for Enzyme Innovation, University of Portsmouth (PORT), Portsmouth, UK. [23]Centre for Genomic Pathogen Surveillance, University of Oxford, Oxford, UK. [24]Clinical Microbiology Department, Queens Medical Centre, Nottingham, UK. [25]Clinical Microbiology, University Hospitals of Leicester NHS Trust, Leicester, UK. [26]County Durham and Darlington NHS Foundation Trust, Darlington, UK. [27]Deep Seq, School of Life Sciences, Queens Medical Centre, University of Nottingham, Nottingham, UK. [28]Department of Infection Biology, Faculty of Infectious & Tropical Diseases, London School of Hygiene & Tropical Medicine, London, UK. [29]Department of Infectious Diseases, King's College London, London, UK. [30]Department of Microbiology, Kettering General Hospital, Kettering, UK. [31]Department of Zoology, University of Oxford, Oxford, UK. [32]Departments of Infectious Diseases and Microbiology, Cambridge University Hospitals NHS Foundation Trust, Cambridge, UK. [33]Division of Virology, Department of Pathology, University of Cambridge, Cambridge, UK. [34]East Kent Hospitals University NHS Foundation Trust, Canterbury, UK. [35]East Suffolk and North Essex NHS Foundation Trust, Colchester, UK. [36]Gateshead Health NHS Foundation Trust, Gateshead, UK. [37]Genomics Innovation Unit, Guy's and St. Thomas' NHS Foundation Trust, London, UK. [38]Gloucestershire Hospitals NHS Foundation Trust, Cheltenham, UK. [39]Great Ormond Street Hospital for Children NHS Foundation Trust, London, UK. [40]Guy's and St. Thomas' BRC, London, UK. [41]Guy's and St. Thomas' Hospitals, London, UK. [42]Hampshire Hospitals NHS Foundation Trust, Basingstoke, UK. [43]Health Data Research UK Cambridge, Cambridge, UK. [44]Health Services Laboratories, London, UK. [45]Heartlands Hospital, Birmingham, UK. [46]Hub for Biotechnology in the Built Environment, Northumbria University, Newcastle, UK. [47]Imperial College Hospitals NHS Trust, London, UK. [48]Institute of Biodiversity, Animal Health & Comparative Medicine, Glasgow, UK. [49]Institute of Microbiology and Infection, University of Birmingham, Birmingham, UK. [50]King's College London, London, UK. [51]Liverpool Clinical Laboratories, Liverpool, UK. [52]Maidstone and Tunbridge Wells NHS Trust, Tunbridge Wells, UK. [53]Manchester University NHS Foundation Trust, Manchester, UK. [54]Microbiology Department, Wye Valley NHS Trust, Hereford, UK. [55]MRC Biostatistics Unit, University of Cambridge, Cambridge, UK. [56]MRC-University of Glasgow Centre for Virus Research, Glasgow, UK. [57]National Infection Service, PHE and Leeds Teaching Hospitals Trust, Leeds, UK. [58]Newcastle Hospitals NHS Foundation Trust, Newcastle, UK. [59]Newcastle University, Newcastle, UK. [60]NHS Greater Glasgow and Clyde, Glasgow, UK. [61]NHS Lothian, Edinburgh, UK. [62]Norfolk and Norwich University Hospital, Norwich, UK. [63]Norfolk County Council, Norwich, UK. [64]North Cumbria Integrated Care NHS Foundation Trust, Carlisle, UK. [65]North Tees and Hartlepool NHS Foundation Trust, Stockton on Tees, UK. [66]Northumbria University, Newcastle, UK. [67]Oxford University Hospitals NHS Foundation Trust, Oxford, UK. [68]PathLinks, Northern Lincolnshire & Goole NHS Foundation Trust, Grimsby, UK. [69]Portsmouth Hospitals University NHS Trust, Portsmouth, UK. [70]Princess Alexandra Hospital Microbiology Dept, Harlow, UK. [71]Public Health Agency, Belfast, UK. [72]Public Health England, London, UK. [73]Public Health England, Clinical Microbiology and Public Health Laboratory, Cambridge, UK. [74]Public Health Scotland, Edinburgh, UK. [75]Public Health Wales NHS Trust, Cardiff, UK. [76]Queen Elizabeth Hospital, London, UK. [77]Queen's University Belfast, Belfast, UK. [78]Royal Devon and Exeter NHS Foundation Trust, Exeter, UK. [79]Royal Free NHS Trust, London, UK. [80]Sandwell and West Birmingham NHS Trust, Birmingham, UK. [81]School of Biological Sciences, University of Portsmouth (PORT), Portsmouth, UK. [82]School of Pharmacy and Biomedical Sciences, University of Portsmouth (PORT), Portsmouth, UK. [83]Sheffield Teaching Hospitals, Sheffield, UK. [84]South Tees Hospitals NHS Foundation Trust, Middlesbrough, UK. [85]Swansea University, Swansea, UK. [86]University Hospitals Southampton NHS Foundation Trust, Southampton, UK. [87]University College London, London, UK. [88]University Hospital Southampton NHS Foundation Trust, Southampton, UK. [89]University Hospitals Coventry and Warwickshire, Coventry, UK. [90]University of Birmingham, Birmingham, UK. [91]University of Birmingham Turnkey Laboratory, Birmingham, UK. [92]University of Brighton, Brighton, UK. [93]University of Cambridge, Cambridge, UK. [94]University of East Anglia, Norwich, UK. [95]University of Edinburgh, Edinburgh, UK. [96]University of Exeter, Exeter, UK. [97]University of Liverpool, Liverpool, UK. [98]University of Sheffield, Sheffield, UK. [99]University of Warwick, Warwick, UK. [100]Viapath, Guy's and St Thomas' NHS Foundation Trust, and King's College Hospital NHS Foundation Trust, London, UK. [101]Virology, School of Life Sciences, Queens Medical Centre, University of Nottingham, Nottingham, UK. [102]Wellcome Centre for Human Genetics, Nuffield Department of Medicine, University of Oxford, Oxford, UK. [103]West of Scotland Specialist Virology Centre, NHS Greater Glasgow and Clyde, Glasgow, UK.

