## [Peer Review File · Nature Communications]

Genomic assessment of quarantine measures to prevent SARS-CoV-2 importation and transmissionREVIEWER COMMENTS

Reviewer #1 (Remarks to the Author):

This is an interesting paper, with a valuable analysis of a good dataset and an important problem. It lacks some clarity however and I think some of the conclusions are not as clear as purported.

Some additional definitions are needed for those who are less familiar with this specific context. There is some additional detail on these in the appendix, but the high-level points should be in the manuscript itself:

“Required” – Please ensure this is the appropriate term and define what it means. Does the policy say it is a “requirement”? Is there any enforcement? In some countries there is, so compliance means something very different from in the UK.

“Imported cases” – How were these identified (e.g., line 80)? Were they tested? Tested based on symptom onset? Symptom onset without a test? I did not see case definitions anywhere. What was the criteria for travel?

“Contacts”- What was the definition of a contact used here? Is there a timing or duration of contact which was used as a definition? Other factors?

For the section on the genomic data, two pieces were unclear. First, was there any location matching? It seems those data were also available (e.g., Fig 4) and would give a lot more insight. Second, I found it hard to interpret these results. They seem almost anecdotal. Is there a reasonable comparison that is possible? Perhaps looking at first isolates that did not have travel history and what their evidence of spreading is would be helpful. There is also a big difference between multiple introductions with potential spread in multiple locations versus few introductions and substantial spread. It seems the authors would have enough data to show something about this, but I didn't see it.

There is an interesting finding of decreased contacts over time for both travelers with and without restrictions (line 124 and Fig 3). Please discuss why this might be the case. It seems there is some other important piece missing.

Lines 171-173: I don't follow the first clause of this sentence; maybe more detail would help. More importantly, the comparison of 5 or more contacts to none seems like a pretty stark comparison. Maybe adding some context here would help. Was there a difference between 1-4 and 5+? Are there sufficient numbers to compare zero to 5+ and have it be meaningful?

Were there any changes in surveillance over time that may have impacted the results? This could be surveillance for imported cases, contact tracing efforts (e.g., Fig 3C?), or sequencing. All of these could impact the results substantially and I didn't see any description of changes in those systems.

Regarding the discussion about the role of London, I think there are a few other components to be considered. The age distribution may be one, but also the presence of multiple airports, businesses, the UK government, a more diverse population, etc. could also be reasons for this. Moreover, it is a large population. So even a high number of importations may have had little impact relative to all of the transmission that was ongoing. One thing that may help clarify this for the reader would be reporting the relative proportion of cases with a traveler history to those who do not.

Line 261: there is a clear intention here to make policy recommendations based on the findings. However, those recommendations are not clear. I think it would be helpful to be more explicit. Specifically, what are the measures with evidence from this study that will likely help? Also be careful about that though. Line 283 says that a 14d quarantine was effective, but would it have

been equally effective if it were 7 days or 10 days? There is no evidence of that here, but evidence from other research that suggests that would likely be as effective and perhaps even more so if compliance were higher. Be clear and circumspect with the recommendations.

Much of the manuscript could use some editing for clarity. There are words missing or misplaced here and there throughout.

Minor suggestions:

Line 78: cite the policy "aim". If not citable, the purpose of the policy is conjecture and the sentence should be rewritten as such.

Line 82: was this compliance specifically for quarantine? State that one way or the other so that it is clear to the reader.

Line 110: What is "duration of the peak". This seems like a specific metric, but I do not see it defined nor characterized in a way that supports this sentence.

Line 123: Which covariates specifically? Please state them rather than "all".

Line 132: define "sufficiently unique" (and better not to start a sentence with a number).

Lines 156-157: not very clear as written. I think the point is that there was no significant difference between the different contact numbers, but the average number was higher with more contacts? Was this controlled for recommendations for the location-time?

Line 229: define R and state what the estimate values were.

Line 248: this says "isolation", but I think refers to "quarantine"?

Line 266-267: I didn't understand this sentence.

Fig 1: the legend is out of order.

Write out dates. They notation used is country specific and can be difficult for others to read.

Reviewer #2 (Remarks to the Author):

This is a very interesting paper describing the effectiveness of quarantine in reducing transmission after travel using phylodynamic approaches. This is a unique dataset in that travel-related SARS-CoV-2 cases and their contacts (identified via contact tracing - test and trace) and their associated genomes (via COG) were analysed. Quarantining reduced contacts. The number of linked cases via genomics was also reduced in those who returned from countries where quarantining was required.

The journal house style seems to differ from this paper. The style is rather jumbled - consider adding sections where appropriate.

Abstract

Line 42-43 Rogue full stop.

Line 43 - spell out C.I on first use

Line 43 attenuation of what? Presumably contacts -

Line 43 Report the attenuated number of contacts

Line 44 - report the rate ratio alone as 40% reduction is repetition

Line 45 - is this a sub-group analysis? If yes then what is the N for high quality SARS-CoV-2 genomes

Line 48 - is this related to one cluster - this seems to be new information relating to contact tracing and should be removed or expanded.

Line 50- For the conclusion - suggest that this study as found an association with 14-day quarantine and reduction onward transmission . Efficacy usually refers to a study of a controlled study rather than real-world.

In the introduction Lines 93-97 an overarching result or conclusion is included rather than an Aim or is this previous work- consider moving to the appropriate section or provide a citation.

When describing the countries with the highest numbers of imports, the duration of the peak imported cases should be described here (line 109).

For geographical variations London is likely to have the highest numbers - the use of a denominator here would be useful to understand representation of England population (geographical area total census population) line 111. The data introduced in line 228 would be helpful.

Where the median reported contacts are reported - line 114 - the inclusion of time periods is important. Were all test positive travel cases contacts followed up over similar time period - or did contact tracing commence later for those in quarantine? If the time periods of when contact tracing began and ended are available it would provide reassurance that quarantine affected number of contacts rather than differential ascertainment of contact tracing

In the summary of findings in the Discussion line 212 - it is worth listing the three countries which accounted for the most imported cases.

The authors recognise the limitations of the COG database with relative limited proportion of full genomic sequencing. Important future work should look to sequence a higher proportion of positive tests to support important surveillance work (such as this study).

In the limitations - it states that the number of contacts were self-reported. Please clarify the role of Test and Trace (T&T) -e.g. in supporting the accuracy of this number.Line 276

What proportion of the non-quarantine cases self-isolated due to positive tests. How would self-isolation post positive test affect the study results.

Minor: Line 212 repeated '%s

Reviewer #3 (Remarks to the Author):

Using the COG-UK sequence database, the authors estimate the proportion of imported covid-19 cases to England in the summer of 2020, and the effects of those cases on onwards transmission. They find that the requirement or not to quarantine on arrival in England had a noticeable effect on the number of onwards cases, particularly in young people, highlighting the importance of travel quarantine for limiting covid-19 transmission.

The findings are intuitive but important; quantifying these effects is paramount for policy making. The analysis seems robust and is clearly explained. My only concern for the study's impact is around timeliness: for policy making, it is hard to extrapolate from the limited study period of summer 2020 to future scenarios, particularly in the face of new transmission dynamics (new

variants, vaccines, differing social restrictions, etc.). If it were possible to update the study with more recent data that would greatly increase the impact, but I appreciate there may be many reasons why that's not possible. Hopefully the work will inspire further follow-up research into these important questions.

I recommend the paper for publication after some minor revisions.

Queries:

Lines 108-109: can I check, is it that 2.9% of the cases in your study which you identified as travel-related came from before 4th July? Rather than (as I first read it) 2.9% of cases between 27th May and 4th July were travel-related?

132-135: Could you comment on the coverage of the COG-UK dataset? It would be nice here to point to Extended Data Figure 12, which I found helpful but only saw later. Could you convert that information into a useful statistic here, for example, if an isolate is >3 SNPs from any sequence in the UK dataset, what's the probability that it is present elsewhere in the UK and just hasn't been sequenced?

248: trend towards positive correlation - can you be more specific? R^2 ?

266-267: Not sure I quite understand this sentence - possibly a grammar issue, and I'm not sure what you mean by "aligned". Throughout this paragraph I got a bit confused about the dates involved and how they relate to each other.

Figure 2: I spent a while trying to understand this plot. Perhaps just a personal thing, but I would find it more intuitive to highlight the times with travel restrictions, rather than without. It appears that travel restrictions had a rapid effect on reducing the number of cases coming from Greece, but the effect of travel restrictions is less / not at all apparent in b-d. I would find the story easier to immediately interpret if all the y-axes had the same range.

Extended data figure 4: what is the red circle?

Notes:

I recommend double-checking the use of "UK" and "England" throughout - there were a couple of times when I wasn't sure whether the right one was being used.

A broad comment that many of the figures in the extended data are nicer (better use of colour, etc.) than in the main text. Many figures would benefit from a larger font size.

There is a wealth of information in the extended figures and tables which answered various questions which came to my mind as I was reading; I would suggest that more pointers could be used throughout the text to signal the availability of this information.

Line 111: Geographically -> Geographical ?

115: is that 172 correct, during a time of restrictions on gathering?

118: require -> required

129: was -> were

212: extra "%" sign

221: destination -> destinations?

243: rogue comma

14th September 2021

We would like to extend our gratitude to the editor and reviewers for their time. We hope to have adequately responded to the highly insightful and helpful comments provided. Please find point-by-point author responses below.

With thanks,

Dr Dinesh Aggarwal

Dr Ewan Harrison
(for the authors)

Reviewer 1	
This is an interesting paper, with a valuable analysis of a good dataset and an important problem. It lacks some clarity however and I think some of the conclusions are not as clear as purported.	Thank you for pointing out the interest generated and value added from this paper. We hope we have addressed your comments adequately to improve the paper.
Some additional definitions are needed for those who are less familiar with this specific context. There is some additional detail on these in the appendix, but the high-level points should be in the manuscript itself:	Thank you for this comment. We have addressed each definition specifically below.
“Required” – Please ensure this is the appropriate term and define what it means. Does the policy say it is a “requirement”? Is there any enforcement? In some countries there is, so compliance means something very different from in the UK.	Thank you for highlighting this important point that is necessary to generalise findings to a global readership. We have provided background to the enforcement policy in England over the study period in lines 423-429.
“Imported cases” – How were these identified (e.g., line 80)? Were they tested? Tested based on symptom onset? Symptom onset without a test? I did not see case definitions anywhere. What was the criteria for travel?	We thank Reviewer 1 for highlighting this. Definitions for ‘imported cases’ are found under ‘Contact tracing and case identification’ (line 395).
“Contacts”- What was the definition of a contact used here? Is there a timing or duration of contact which was used as a definition? Other factors?	Thank you very much for highlighting this omission. We have provided additional definitions in the text under ‘Contact tracing and case identification’ (lines 413-415).
For the section on the genomic data, two pieces were unclear. First, was there any location matching? It seems those data were also available (e.g., Fig 4) and would give a lot more insight. Second, I found it hard to interpret these results. They seem almost	We thank Reviewer 1 for this comment. Addressing the first question (location matching): the purpose of this analysis is to specifically corroborate epidemiological findings and understand the efficacy of a non-pharmacological intervention (travel-related quarantine) and not to assess the introduction and transmission of clusters from specific countries. We do

anecdotal. Is there a reasonable comparison that is possible? Perhaps looking at first isolates that did not have travel history and what their evidence of spreading is would be helpful. There is also a big difference between multiple introductions with potential spread in multiple locations versus few introductions and substantial spread. It seems the authors would have enough data to show something about this, but I didn't see it.

however provide a table of destination countries from which genomes were available (Supplementary Table 4) and a Figure representing location of travellers within the UK (Fig. 1e) to provide insight of where these genomes had originated from (including unique genomes). Further, we use destination country as a covariate in the model to assess the impact of quarantine on contacts per case.

Addressing question two: we are using the unique nature of our large epidemiological dataset and imported SARS-CoV-2 genomes to enable us to define clusters that can be used to assess an important non-pharmacological intervention (travel restriction) based on whether they belong to the travel-related 'quarantine' or 'no quarantine' cohort. We are not defining clusters of non-travel related clusters which would encounter the issue of limited SARS-CoV-2 genomic diversity (our study overcomes this by specifically choosing those genomes that are unique). Selecting clusters based on the code we have written (utilising the benefit of identifying unique index case genomes) encounters an issue when applied to non-travel genomes – we cannot be certain these unique cases with long stem-lengths are certainly *not* travel-related genomes.

We do however completely agree that providing context to data is important and thank the reviewer for highlighting this. We have used the entire dataset of SARS-CoV-2 cases (105,794 non-travel related cases who reported 233,182 contacts) in England for the study period to provide context to the epidemiological analysis when attempting to shed further light on the reduction in contacts per case over time. Here, we demonstrate the contacts per case for non-travellers to be consistently lower over the study period than the travel-related cohort.

Addressing question three regarding multiple introductions: we thank Reviewer 1 for highlighting this to us and we not only agree this is an important consideration but by assessing multiple introductions of a genome within unique clusters, we have provided significant insight into the nature of the effect of travel-related quarantine on cluster size. We have used our unique travel-related dataset to account for cluster size according to the number of introductions (lines

	219-244).
There is an interesting finding of decreased contacts over time for both travelers with and without restrictions (line 124 and Fig 3). Please discuss why this might be the case. It seems there is some other important piece missing.	We thank the reviewer for pointing this out. Indeed, there is an interesting finding of decreased contacts over time which would benefit from further investigation. We have attempted to understand if this effect is seen independent of travel by assessing non-travel cases during the study period (line 164-173). We found that the time-related reduction in contacts per case is specific to travel-related SARS-CoV-2 cases, which would require additional data on traveller type and behaviour to evaluate further (discussed lines 329-340). Unfortunately, we don't have access to this data so are unable to provide any further insights.
Lines 171-173: I don't follow the first clause of this sentence; maybe more detail would help. More importantly, the comparison of 5 or more contacts to none seems like a pretty stark comparison. Maybe adding some context here would help. Was there a difference between 1-4 and 5+? Are there sufficient numbers to compare zero to 5+ and have it be meaningful?	Thank you for highlighting the issues with this comparison. We agree the manner in which this work has been described is confusing and we have therefore simplified representation of association between contacts per index case and cluster size (Supplementary Fig. 1 and lines 236-244). Again, our work on multiple introductions largely explains these findings.
Were there any changes in surveillance over time that may have impacted the results? This could be surveillance for imported cases, contact tracing efforts (e.g., Fig 3C?), or sequencing. All of these could impact the results substantially and I didn't see any description of changes in those systems.	We thank Reviewer 1 for this highly insightful comment. We have provided further information to reflect a lack of significant change in the contact tracing effort (Supplementary Fig. 13). We can confirm there were no significant changes in surveillance of import cases other than that previously stated (lines 399-402) over study period. Travel policy changes were specifically related to quarantine requirements, which is the basis of this paper. We believe the above further underlines the reliability of this study.
Regarding the discussion about the role of London, I think there are a few other components to be considered. The age distribution may be one, but also the presence of multiple airports, businesses, the UK government, a more diverse population, etc. could also be reasons for this. Moreover, it is a large population. So even a high number of importations may have had little impact relative to all of the	We thank Reviewer 1 for this comment. We agree this aspect of the discussion requires further information and have provided this accordingly (lines 311-319).

transmission that was ongoing. One thing that may help clarify this for the reader would be reporting the relative proportion of cases with a traveler history to those who do not.	
Line 261: there is a clear intention here to make policy recommendations based on the findings. However, those recommendations are not clear. I think it would be helpful to be more explicit. Specifically, what are the measures with evidence from this study that will likely help? Also be careful about that though. Line 283 says that a 14d quarantine was effective, but would it have been equally effective if it were 7 days or 10 days? There is no evidence of that here, but evidence from other research that suggests that would likely be as effective and perhaps even more so if compliance were higher. Be clear and circumspect with the recommendations.	We thank Reviewer 1 for this highly thoughtful comment. We have been careful to not overstate our conclusions. We have specifically drawn conclusions from the period of time this study covers and policies implemented. The purpose of this paper is to assess/quantify the efficacy of a highly impactful and restrictive travel policy that is employed globally (albeit different iterations of the same policy) and therefore better inform its implementation. We also clearly demonstrate and express the need to gather and analyse detailed epidemiological and genomic surveillance data. We agree that the discussion would benefit from clarity and have adjusted the discussion in multiple areas to make it clear where public health interventions related to our findings would help. The study period does not evaluate the use of a 7 day or 10 day quarantine, and therefore we draw conclusions based on a 14 day quarantine. We do fully agree that policy making should consider more recent evidence that would support a similar efficacy with a shorter quarantine period (https://www.cdc.gov/coronavirus/2019-ncov/science/science-briefs/scientific-brief-options-to-reduce-quarantine.html). We have therefore inserted a comment in the limitations to reflect this (lines 372-373).
Much of the manuscript could use some editing for clarity. There are words missing or misplaced here and there throughout.	We thank Reviewer 1 for highlighting this and have edited the manuscript accordingly.
Minor suggestions:	
Line 78: cite the policy “aim”. If not citable, the purpose of the policy is conjecture and the sentence should be rewritten as such.	Thank you for pointing this out, we have appropriately cited and re-worded aspects that are conjecture (lines 80-82).

Line 82: was this compliance specifically for quarantine? State that one way or the other so that it is clear to the reader.	Thank you for highlighting this. This sentence refers specifically to travel-related quarantine and we have clarified this in the text (lines 84-85).
Line 110: What is “duration of the peak”. This seems like a specific metric, but I do not see it defined nor characterized in a way that supports this sentence.	We thank Reviewer 1 for this comment; the purpose of this figure is purely to describe and display the data for readers, it is not to draw significant conclusions, which should be through the subsequent analysis. To avoid ambiguity, we have deleted this sentence (line 128).
Line 123: Which covariates specifically? Please state them rather than “all”.	Thank you for this comment, we have clarified in lines 144-145 and 149-150.
Line 132: define “sufficiently unique” (and better not to start a sentence with a number).	We have provided a definition in the text and additionally restructured the sentence (lines 184-186).
Lines 156-157: not very clear as written. I think the point is that there was no significant difference between the different contact numbers, but the average number was higher with more contacts? Was this controlled for recommendations for the location-time?	Thank you for highlighting the issues with this comparison. We agree the manner in which this work has been described is confusing and we have therefore simplified representation of association between contacts per index case and cluster size (Supplementary Fig. 1 and lines 236-244). Again, our work on multiple introductions largely explains these findings.
Line 229: define R and state what the estimate values were.	Thank you for highlighting this, we have defined and stated estimate values as suggested (lines 316-318).
Line 248: this says “isolation”, but I think refers to “quarantine”?	Thank you, we have largely replaced this paragraph to reflect addition analysis including this sentence.
Line 266-267: I didn’t understand this sentence.	Thank you for highlighting the ambiguity here, we have altered this sentence to better reflect the limitation (line 367-369).
Fig 1: the legend is out of order.	Thank you very much for pointing this out, we have corrected this.
Write out dates. They notation used is country specific and can be difficult for others to read.	Thank you for highlighting this, we have ensured all dates are written out in the text as suggested.

Reviewer 2	
This is a very interesting paper describing the effectiveness of quarantine in reducing transmission after travel using phylodynamic approaches. This is a unique dataset in that travel-related SARS-CoV-2 cases and their contacts (identified via contact tracing - test and trace) and their associated genomes (via COG) were analysed. Quarantining reduced contacts. The number of linked cases via genomics was also reduced in those who returned from countries where quarantining was required.	Thank you for this comment and we are glad that the manuscript was found to be very interesting. We agree this is a highly unique large dataset reflecting a period of time where fewer additional factors such as vaccination and variants of concern allowed for more reliable evaluation of policy effects such as travel-related quarantine.
The journal house style seems to differ from this paper. The style is rather jumbled - consider adding sections where appropriate.	We thank Reviewer 2 for this comment and have added appropriate sections to the manuscript.
Line 42-43 Rogue full stop.	We thank Reviewer to for this comment and have removed this full stop.
Line 43 - spell out C.I on first use	We thank Reviewer to for this comment and have spelt out C.I on first use (line 43).
Line 43 attenuation of what? Presumably contacts	Thank you for pointing this out, we have provided clarification in the text (line 44).
Line 43 Report the attenuated number of contacts	Thank you for highlighting this, we have provided this value in the text (line 44).
Line 44 - report the rate ratio alone as 40% reduction is repetition	We agree with this comment and have adjusted the abstract accordingly (line 45).
Line 45 - is this a sub-group analysis? If yes then what is the N for high quality SARS-CoV-2 genomes	Thank you for pointing this out, we have provided clarification in the text (line 46).
Line 48 - is this related to one cluster - this seems to be new information relating to contact - tracing and should be removed or	Thank you for pointing this out, we have provided clarification in the text (lines 53-54).

expanded.	
Line 50- For the conclusion - suggest that this study as found an association with 14-day quarantine and reduction onward transmission . Efficacy usually refers to a study of a controlled study rather than real-world.	We thank Reviewer 2 for providing this comment. We have adjusted the conclusion to better reflect the analysis conducted (lines 55-57).
In the introduction Lines 93-97 an overarching result or conclusion is included rather than an Aim or is this previous work- consider moving to the appropriate section or provide a citation.	We thank Reviewer 2 for highlighting this comment. We have deleted this text accordingly.
When describing the countries with the highest numbers of imports, the duration of the peak imported cases should be described here (line 109).	We thank Reviewer 2 for this comment; the purpose of this figure is purely to describe and display the data for readers, it is not to draw significant conclusions, which should be through the subsequent analysis. To avoid ambiguity, we have deleted this sentence (line 128).
For geographical variations London is likely to have the highest numbers - the use of a denominator here would be useful to understand representation of England population (geographical area total census population) line 111. The data introduced in line 228 would be helpful.	We thank Reviewer 2 for this comment. We have provided this information (line 131) and additionally provided further clarity to this finding in the discussion, including a denominator for the extent of infections specific to London for context (lines 311-315).
Where the median reported contacts are reported - line 114 - the inclusion of time periods is important. Were all test positive travel cases contacts followed up over similar time period - or did contact tracing commence later for those in quarantine? If the time periods of when contact tracing began and ended are available it would provide reassurance that quarantine affected number of contacts rather than differential ascertainment of contact tracing	Contact tracing was consistent for SARS-CoV-2 cases regardless of quarantine status (defined in lines 404-411). All contacts were ascertained for cases in the 2 days prior to symptom onset or test date (whichever earlier) and continued until contact tracing system completion, but with the understanding that cases would not be expected to have further contacts as they would be in isolation from the point of symptom onset or a positive test.
In the summary of findings in the	We thank Reviewer 2 for highlighting this and have

Discussion line 212 - it is worth listing the three countries which accounted for the most imported cases.	added this information in the discussion (line 292)
The authors recognise the limitations of the COG database with relative limited proportion of full genomic sequencing. Important future work should look to sequence a higher proportion of positive tests to support important surveillance work (such as this study).	We thank Reviewer 2 for this comment and we continue to highlight the need for adequate sequencing at a global level.
In the limitations - it states that the number of contacts were self-reported. Please clarify the role of Test and Trace (T&T) -e.g. in supporting the accuracy of this number.Line 276	Reporting bias is mitigated through mandatory completion of a passenger locator form to assist identification of returning travellers. Positive cases are contacted by Test and Trace via online or call centre tracing. Additionally travel-related cases are seen as higher risk and therefore referred to local public health agencies for targeted contact tracing (additional information provided in Supplementary Methods and Definitions).
What proportion of the non-quarantine cases self-isolated due to positive tests. How would self-isolation post positive test affect the study results.	We thank Reviewer 2 for this comment. All cases here are positive and therefore should be self-isolating regardless of quarantine status. Contacts are 2 days prior to onset of symptoms/date of test and this is being evaluated (here, reported to be lower for those quarantining vs those not), whilst genomic evaluation is specifically of clusters generated by index cases related to open vs closed travel corridor. We expect self-isolation after a positive test result would affect results for quarantine and non-quarantine equally.
Minor: Line 212 repeated '%s	We thank Reviewer 2 for highlighting this and we have corrected the text.
Reviewer 3	
Using the COG-UK sequence database, the authors estimate the proportion of imported covid-19 cases to England in the summer of 2020, and the effects of those cases on onwards transmission. They find that the requirement or not to quarantine on arrival in England had a noticeable effect on	We thank Reviewer 3 for highlighting the importance of the results we present. Regarding the request to provide more recent data, we are working with public health agencies to implement this internally for real time analysis. This dataset however represents a unique period prior to emergence of variants of concern and roll out of vaccination. This allows us to evaluate the specific effect of travel restriction in a

the number of onwards cases, particularly in young people, highlighting the importance of travel quarantine for limiting covid-19 transmission. The findings are intuitive but important; quantifying these effects is paramount for policy making. The analysis seems robust and is clearly explained. My only concern for the study's impact is around timeliness: for policy making, it is hard to extrapolate from the limited study period of summer 2020 to future scenarios, particularly in the face of new transmission dynamics (new variants, vaccines, differing social restrictions, etc.). If it were possible to update the study with more recent data that would greatly increase the impact, but I appreciate there may be many reasons why that's not possible. Hopefully the work will inspire further follow-up research into these important questions. I recommend the paper for publication after some minor revisions.	more reliable manner (fewer competing variables). At this point in time, there was no specific difference noted in transmissibility of different lineages, making behavioural changes (such as those secondary to travel policy and evaluated here) a dominant force in mitigating or facilitating transmission, as demonstrated by Hodcroft et al whilst evaluating the transmission of SARS-CoV-2 a222v mutants (Nature, 2021).
Lines 108-109: can I check, is it that 2.9% of the cases in your study which you identified as travel-related came from before 4th July? Rather than (as I first read it) 2.9% of cases between 27th May and 4th July were travel-related?	We thank Reviewer 3 for pointing out the ambiguity in the text. 2.9% of the cases in the study were identified as travel-related came from before the 4th July. We have clarified in this in the manuscript (lines 126-127).
132-135: Could you comment on the coverage of the COG-UK dataset? It would be nice here to point to Extended Data Figure 12, which I found helpful but only saw later. Could you convert that information into a useful statistic here, for example, if an isolate is >3 SNPs from any sequence in the UK dataset, what's the probability that it is present elsewhere in the UK and just hasn't been sequenced?	We thank Reviewer 3 for this comment. While we agree that bringing in coverage would be useful, such a calculation of the probably would be highly complex, as it would need to consider both local and national coverage at any one time and for each case. By using SNP distance of >3 SNPS we aim to improve certainty that our known imported SARS-CoV-2 cases are truly unique. We firstly know the expected rate of mutation of SARS-CoV-2 to be ~2 SNPs over 4 weeks but allow for a 4 SNP difference; this not only includes a 4 SNP difference from a previously sampled COG-UK

	genome but also any COG-UK genome that is within 4 SNPs but is part of separate transmission chains. We agree there is a chance of potentially missed transmission chains through under-sampling, and we have accordingly inserted a sentence in the limitations section to help make this clear to readers (lines 364-366).
248: trend towards positive correlation - can you be more specific? R ² ?	We thank Reviewer 3 for highlighting this. We have deleted this sentence from the manuscript.
266-267: Not sure I quite understand this sentence - possibly a grammar issue, and I'm not sure what you mean by "aligned". Throughout this paragraph I got a bit confused about the dates involved and how they relate to each other.	Thank you for highlighting the ambiguity here, we have altered this sentence to better reflect the limitation (lines 367-369).
Figure 2: I spent a while trying to understand this plot. Perhaps just a personal thing, but I would find it more intuitive to highlight the times with travel restrictions, rather than without. It appears that travel restrictions had a rapid effect on reducing the number of cases coming from Greece, but the effect of travel restrictions is less / not at all apparent in b-d. I would find the story easier to immediately interpret if all the y-axes had the same range.	We agree with Reviewer 3 and thank them for this comment; we have adjusted Figure 2 to highlight times with travel restrictions and adjusted y-axes to remain consistent between countries.
Extended data figure 4: what is the red circle?	We thank the reviewer for pointing this out – this is a mistake and has been removed.
Notes:	
I recommend double-checking the use of "UK" and "England" throughout - there were a couple of times when I wasn't sure whether the right one was being used.	We thank Reviewer 3 for highlighting this and have addressed any ambiguity in the text.
A broad comment that many of the	We thank Reviewer 3 for highlighting this comment.

figures in the extended data are nicer (better use of colour, etc.) than in the main text. Many figures would benefit from a larger font size.	We have specifically chosen limited colours to represent data in the main manuscript to support individuals with colour blindness. We can further adjust font size if required.
There is a wealth of information in the extended figures and tables which answered various questions which came to my mind as I was reading; I would suggest that more pointers could be used throughout the text to signal the availability of this information.	We thank Reviewer 3 for highlighting the substantial amount of additional data and analysis presented in the Supplementary files. We have attempted to better sign-post these in the text.
Line 111: Geographically -> Geographical ?	Thank you for highlighting this. We have corrected the text accordingly.
115: is that 172 correct, during a time of restrictions on gathering?	This is indeed the correct number of contacts. From extended metadata we can see the majority of these contacts are encountered during a return flight. We have included additional work to assess the robustness of our conclusions when top-coding cases with greater than 10 contacts as having '10 contacts'; we demonstrate no change in our conclusions (lines 151-156).
118: require -> required	We thank Reviewer 3 for pointing this out. We have adjusted the text accordingly.
129: was -> were	We thank Reviewer 3 for pointing this out. We have adjusted the text accordingly.
212: extra “%” sign	We thank Reviewer 3 for pointing this out. We have adjusted the text accordingly.
221: destination -> destinations?	We thank Reviewer 3 for pointing this out. We have adjusted the text accordingly.
243: rogue comma	We thank Reviewer 3 for pointing this out. We have adjusted the text accordingly.

REVIEWERS' COMMENTS

Reviewer #2 (Remarks to the Author):

I am happy with the revisions.

Reviewer #3 (Remarks to the Author):

Thank you to the authors for thoroughly addressing all of my concerns and comments. I recommend the paper for publication.